# Integrated view and comparative analysis of baseline protein expression in mouse and rat tissues

**Shengbo Wang**[1‡], **David García-Seisdedos**[1,2‡], **Ananth Prakash**[1,2‡*], **Deepti Jaiswal Kundu**[1], **Andrew Collins**[3], **Nancy George**[1], **Silvie Fexova**[1], **Pablo Moreno**[1], **Irene Papatheodorou**[1,2], **Andrew R. Jones**[3], **Juan Antonio Vizcaíno**[1,2*]

**1** European Molecular Biology Laboratory - European Bioinformatics Institute (EMBL-EBI), Wellcome Genome Campus, Hinxton, Cambridge, United Kingdom, **2** Open Targets, Wellcome Genome Campus, Hinxton, Cambridge, United Kingdom, **3** Institute of Systems, Molecular and Integrative Biology, University of Liverpool, Liverpool, United Kingdom

‡ All three authors have contributed equally and they wish to be considered as joint first authors.
* ananth@ebi.ac.uk (AP); juan@ebi.ac.uk (JAV)

**Data Availability Statement:** The R code used for downstream data analyses and for plotting of results presented in this manuscript are available

## Abstract

The increasingly large amount of proteomics data in the public domain enables, among other applications, the combined analyses of datasets to create comparative protein expression maps covering different organisms and different biological conditions. Here we have reanalysed public proteomics datasets from mouse and rat tissues (14 and 9 datasets, respectively), to assess baseline protein abundance. Overall, the aggregated dataset contained 23 individual datasets, including a total of 211 samples coming from 34 different tissues across 14 organs, comprising 9 mouse and 3 rat strains, respectively.

In all cases, we studied the distribution of canonical proteins between the different organs. The number of canonical proteins per dataset ranged from 273 (tendon) and 9,715 (liver) in mouse, and from 101 (tendon) and 6,130 (kidney) in rat. Then, we studied how protein abundances compared across different datasets and organs for both species. As a key point we carried out a comparative analysis of protein expression between mouse, rat and human tissues. We observed a high level of correlation of protein expression among orthologs between all three species in brain, kidney, heart and liver samples, whereas the correlation of protein expression was generally slightly lower between organs within the same species. Protein expression results have been integrated into the resource Expression Atlas for widespread dissemination.

## Author summary

We have reanalysed 23 baseline mass spectrometry-based public proteomics datasets stored in the PRIDE database. Overall, the aggregated dataset contained 211 samples, coming from 34 different tissues across 14 organs, comprising 9 mouse and 3 rat strains, respectively. We analysed the distribution of protein expression across organs in both

from https://github.com/Ananth-Prakash/Mouse_Rat_DDA_proteomics.

**Funding:** AP and DGS were funded by Open Targets (project OTAR-02-068), https://www.opentargets.org/ AP, IP, NG, PM, SF and JAV were funded by BBSRC BB/T019670/1, https://bbsrc.ukri.org/ AC and ARJ were funded by BBSRC BB/T019557/1, https://bbsrc.ukri.org/ AP, DGS, SW and JAV were funded by Wellcome Trust [grant number 208391/Z/17/Z], https://wellcome.org/ DJK, IP and JAV were funded by EMBL core funding, https://www.embl.org/. The funders had no role in study design, data collection and analysis, decision to publish, or preparation of the manuscript.

**Competing interests:** The authors have declared that no competing interests exist.

species. We also studied how protein abundances compared across different datasets and organs for both species. Then we performed gene ontology and pathway enrichment analyses to identify enriched biological processes and pathways across organs. We also carried out a comparative analysis of baseline protein expression across mouse, rat and human tissues, observing a high level of expression correlation among orthologs in all three species, in brain, kidney, heart and liver samples. To disseminate these findings, we have integrated the protein expression results into the resource Expression Atlas.

## 1. Introduction

High-throughput mass spectrometry (MS)-based proteomics approaches have matured significantly in recent years, becoming an essential tool in biological research [1]. This has been the consequence of very significant technical improvements in MS instrumentation, chromatography, automation in sample preparation and computational analyses, among other areas. The most used MS-based experimental approach is Data Dependent Acquisition (DDA) bottom-up proteomics. Among the main quantitative proteomics DDA techniques, label-free intensity-based approaches remain very popular, although labelled-approaches, especially those techniques based on the isotopic labelling of peptides ($MS^2$ labelling), such as iTRAQ (Isobaric tag for relative and absolute quantitation) and TMT (Tandem Mass Tagging), are becoming increasingly used as well.

Following the steps initiated by genomics and transcriptomics, open data practices in the field have become embedded and commonplace in proteomics in recent years. In this context, datasets are now commonly available in the public domain to support the claims published in the corresponding manuscripts. The PRIDE database [2], located at the European Bioinformatics Institute (EBI), is currently the largest resource worldwide for public proteomics data deposition. PRIDE is also one of the founding members of the global ProteomeXchange consortium [3], involving five other resources, namely PeptideAtlas, MassIVE, iProX, jPOST and PanoramaPublic. ProteomeXchange has standardised data submission and dissemination of public proteomics data worldwide.

As a consequence, there is an unprecedented availability of data in the public domain, which is triggering multiple applications [4], including the joint reanalysis of datasets (so-called meta-analysis studies) [5–7]. Indeed, public proteomics datasets can be systematically reanalysed and integrated e.g., to confirm the results reported in the original publications, potentially in a more robust manner since evidence can be strengthened if it is found consistently across different datasets. Potentially, new insights different to the aims of the original studies can also be obtained by reanalysing the datasets using different strategies, this includes repurposing of public datasets [8], including for instance approaches such as proteogenomics studies for genome annotation purposes [9–12].

In this context of reuse of public proteomics data, PRIDE has started to work on developing data dissemination and integration pipelines into popular added-value resources at the EBI. This is perceived as a more sustainable approach in the medium-long term than setting up new independent bioinformatics resources. One of them is Expression Atlas [13], a resource that has enabled over the years easy access to gene expression data across species, tissues, cells, experimental conditions and diseases. Only recently, protein expression information coming from reanalysed datasets has been integrated in the 'bulk' section of Expression Atlas. As a result, proteomics expression data can be integrated with transcriptomics information, mostly coming from RNA-Seq experiments. So far, we have performed two meta-analysis studies

involving the reanalysis and integration of: (i) 11 public quantitative datasets coming from cell lines and human tumour samples [13]; and (ii) 24 human baseline datasets coming from 31 different organs [14].

The next logical step is to perform an analogous study of baseline protein expression in two of the main model organisms: *Mus musculus* and *Rattus norvegicus*. To date, there are only a small number of bioinformatics resources providing access to reanalysed MS-based quantitative proteomics datasets, and even fewer if one considers only mouse and rat data. In this context, at the end of 2020, ProteomicsDB [15] released a first version of the mouse proteome, based on the reanalysis of five label-free datasets. To the best of our knowledge, there is no such public resource storing accurate MS-derived data for rat data yet. PaxDB is a resource [16] that provides protein expression information coming from many species (including mouse and rat) but the reported data relies on spectral counting, a technique that generally does not provide the same level of accuracy than intensity-based label-free approaches. Additionally, although antibody-based human protein expression information is provided *via* the Human Protein Atlas [17], their efforts are focused on human protein expression.

Here, we report the reanalysis and integration of 23 public mouse (14 datasets) and rat (9 datasets) label-free datasets, and the incorporation of the results into the resource Expression Atlas as baseline studies. Additionally, we report a comparative analysis of protein expression across mouse, rat and human (in this case using the results reported at [14] using the same methodology).

## 2. Results

### 2.1. Baseline proteomics datasets

Overall, we quantified protein expression from 34 healthy tissues in 14 organs coming from 23 datasets. The analyses covered a total of 1,173 MS runs from 211 samples that were annotated as healthy/control/non-treated samples, thus representing baseline protein expression. Non-control/disease samples associated with these datasets were also reanalysed but are not discussed here. Normalised protein abundances values (as ppb, parts per billion, see Methods for calculation) from both control/healthy/non-treated and disease/treated tissue samples are available to view as heatmaps in Expression Atlas. The protein abundances along with sample annotations, sample quality assessment summary and experimental parameter inputs for Max-Quant can be downloaded from Expression Atlas as text files. A summary of the data selection and reanalysis protocols is shown in Fig 1. The total number of peptides and proteins identified in these datasets are shown in Table 1.

### 2.2. Protein coverage across organs and datasets

One of our main aims was to study protein expression across various organs. To enable a simpler comparison [14] we first grouped 34 different tissues into 14 distinct organs, as discussed in 'Methods'. We defined 'tissue' as a distinct functional or structural region within an 'organ'. We estimated the number of 'canonical proteins' identified across organs by first mapping all members of each protein group to their respective parent genes. We defined the parent gene as equivalent to the UniProt 'canonical protein' and we will denote the term 'protein abundance' to mean 'canonical protein abundance' from here on in the manuscript.

**2.2.1. Mouse proteome.** A total of 21,274 protein groups were identified from mouse datasets, among which 8,176 protein groups (38.4%) were uniquely present in only one organ and 70 protein groups (0.3%) were ubiquitously observed (see the full list in S2 File). This does not imply that these proteins are unique to these organs. Merely, this is the outcome considering the selected datasets. Mouse protein groups were mapped to 12,570 genes (canonical

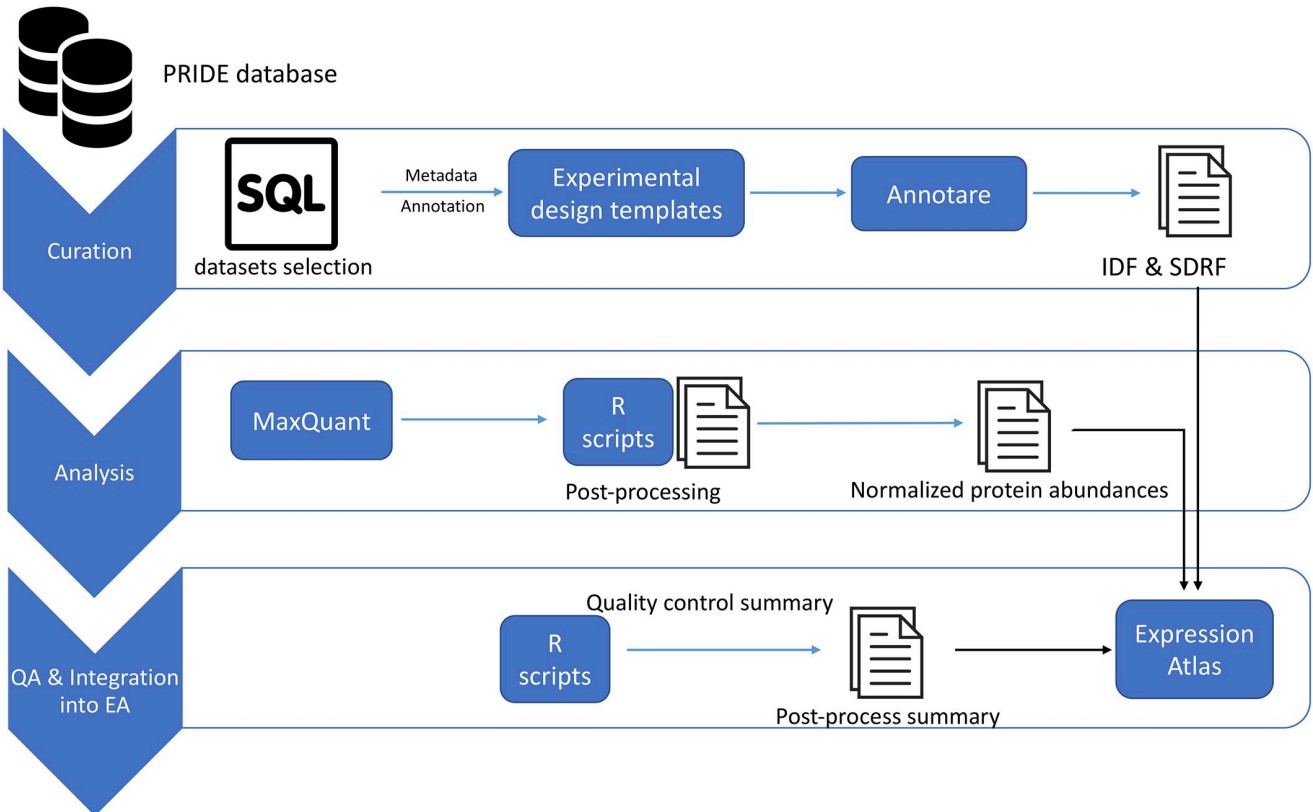

**Fig 1. An overview of the study design and reanalysis pipeline.** QA: Quality assessment.

proteins) (S3 File). We detected the largest number of canonical proteins in samples coming from liver (9,920, 78.9% of the total) and the lowest numbers in samples from tendon (273, 2.2%) and articular cartilage (1,519, 12.1%) (Fig 2A). In the case of tendon and articular cartilage, both experiments did not include sample fractionation in their sample preparation methodology, which can also explain the lower number of detected proteins. The comparatively even lower number of proteins identified in tendon could be attributed to the smallest sample size (only one sample out of 114, 0.9%). Also, tendon is a relatively hypocellular tissue, which has a low protein turnover rate. Dataset PXD000867, containing mouse liver samples, had the highest number of canonical proteins detected (9,715, 77.3%), while the smallest number of proteins was detected in dataset PXD004612 (tendon, 273, 2.2%), as highlighted above (Fig 2C).

We studied the normalised protein abundance distribution in organs (Fig 2B) and found that all organs, except tendon, had similar median abundances. However, one cannot attribute further biological meaning to these observations, since by definition the method of normalisation fixes each sample to have the same "total abundance", which then gets shared out amongst all proteins. The normalised protein abundance distribution in datasets indicated a higher than median abundances detected in datasets PXD004612 (tendon) and PXD003164 (testis) (Fig 2D). A linear relationship was observed between the number of canonical proteins detected in datasets and organs, when compared to the relative amount of their spectral data (Fig 2E). We found a significant number of proteins uniquely detected in one organ (Fig 2F). However, the list of concrete canonical proteins that were detected in just one organ should be

**Table 1. List of mouse and rat proteomics datasets that were reanalysed.**

| Expression Atlas accession numbers | PRIDE dataset identifiers | Tissues | Organs | Species | Strains | Fractionation | Number of MS runs | Number of samples | Number of protein groups[†] | Number of peptides[†] | Number of unique peptides[†] | Number of unique genes mapped[†] |
|---|---|---|---|---|---|---|---|---|---|---|---|---|
| E-PROT-7[§] | PXD000867 [18] | Liver | Liver | *Mus musculus* | C57BL/6J | Yes | 24 | 4 | 12,792 | 246,738 | 167,725 | 9,715 |
| E-PROT-10[§] | PXD000288 [19] | Triceps muscles | Triceps Muscles | *Mus musculus* | C57BL/6 | Yes | 36 | 3 | 10,870 | 189,553 | 126,670 | 6,421 |
| E-PROT-16 | PXD003155 [20] | Cerebellum, Liver | Brain, Liver | *Mus musculus* | C57BL/6 | No | 24 | 12 | 4,508 | 59,696 | 45,728 | 3,797 |
| E-PROT-74 | PXD004612 [21] | Achilles and Plantaris tendon | Tendon | *Mus musculus* | C57BL/6 | No | 8 | 8 | 457 | 6,643 | 3,271 | 273 |
| E-PROT-75 | PXD005230 [22] | Hippocampus, Cerebellum, Cortex | Brain | *Mus musculus* | C57BL/10J | Yes | 72 | 36 | 7,663 | 63,479 | 41,683 | 6,037 |
| E-PROT-76 | PXD009909 [23] | Retina | Eye | *Mus musculus* | ND4 Swiss Webster | Yes | 12 | 1 | 5,002 | 29,454 | 24,961 | 3,686 |
| E-PROT-77 | PXD012307 [24] | Lung | Lung | *Mus musculus* | C57BL/6 | No | 32 | 2 | 6,809 | 106,391 | 73,950 | 5,795 |
| E-PROT-78 | PXD009639 [25] | Lens | Eye | *Mus musculus* | CD1 | Yes | 10 | 1 | 4,519 | 20,779 | 18,006 | 3,064 |
| E-PROT-79 | PXD019394 [26] | Heart, Kidney, Liver, Lung, Brain, Spleen, Testis, Pancreas | Heart, Kidney, Liver, Lung, Brain, Spleen, Testis, Pancreas | *Mus musculus* | Swiss-Webster | Yes | 96 | 8 | 9,853 | 141,506 | 105,701 | 8,185 |
| E-PROT-81 | PXD012636 [27] | Left atrium, Left ventricle, Right atrium, Right ventricle | Heart | *Mus musculus* | C57BL/6 | Yes | 120 | 4 | 7,772 | 146,966 | 99,577 | 6,435 |
| E-PROT-82 | PXD019431 [28] | Articular cartilage | Articular cartilage | *Mus musculus* | BALB\_c | No | 72 | 6 | 1,815 | 17,695 | 15,191 | 1,518 |
| E-PROT-83 | PXD022614 [29] | Brain | Brain | *Mus musculus* | C57BL/6J:Rj C57BL/6JRccHsd | Yes | 120 | 6 | 6,645 | 97,443 | 69,884 | 5,673 |
| E-PROT-84 | PXD004496 [30] | Hippocampus | Brain | *Mus musculus* | C57BL/6J | Yes | 204 | 17 | 4,192 | 37,363 | 30,100 | 3,424 |
| E-PROT-85 | PXD008736 [31] | Right atrium, Sinus node | Heart | *Mus musculus* | C57BL/6J | Yes | 143 | 6 | 7,906 | 144,926 | 94,379 | 6,554 |
| E-PROT-86[§] | PXD012677 [32] | Amygdala | Brain | *Rattus norvegicus* | Sprague Dawley | No | 3 | 3 | 1,872 | 15,326 | 12,367 | 1,382 |
| E-PROT-87[§] | PXD006692 [33] | Lung | Lung | *Rattus norvegicus* | Sprague Dawley | No | 10 | 10 | 2,079 | 14,440 | 11,696 | 1,398 |
| E-PROT-88[§] | PXD016793 [34] | Liver | Liver | *Rattus norvegicus* | Sprague Dawley | No | 8 | 8 | 4,787 | 57,998 | 46,411 | 3,743 |
| E-PROT-89[§] | PXD004364 [35] | Testis | Testis | *Rattus norvegicus* | Sprague Dawley | No | 3 | 3 | 2,351 | 15,880 | 13,674 | 1,601 |
| E-PROT-91 | PXD001839 [36] | Left ventricle | Heart | *Rattus norvegicus* | F344/BN | No | 12 | 12 | 1,345 | 10,310 | 8,804 | 925 |
| E-PROT-92[§] | PXD013543 [37] | Left ventricle | Heart | *Rattus norvegicus* | Wistar | No | 8 | 8 | 1,858 | 17,303 | 13,622 | 1,340 |

*(Continued)*

**Table 1.** (Continued)

| Expression Atlas accession numbers | PRIDE dataset identifiers | Tissues | Organs | Species | Strains | Fractionation | Number of MS runs | Number of samples | Number of protein groups[†] | Number of peptides[†] | Number of unique peptides[†] | Number of unique genes mapped[†] |
|---|---|---|---|---|---|---|---|---|---|---|---|---|
| E-PROT-93 | PXD016958 [38] | First segment of proximal tubule, second segment of proximal tubule, third segment of proximal tubule, medullary thick ascending limb, cortical thick ascending limb, distal convoluted tubule, connecting tubule, cortical collecting duct, outer medullary collecting duct, inner medullary collecting duct | Kidney | *Rattus norvegicus* | Sprague Dawley | Yes | 132 | 32 | 7,846 | 103,886 | 83,662 | 6,130 |
| E-PROT-94 | PXD003375 [39] | Caudal and rostral segments of spinal cord | Spinal cord | *Rattus norvegicus* | Wistar | Yes | 21 | 18 | 2,477 | 29,213 | 22,025 | 1,926 |
| E-PROT-95[§] | PXD015928 [40] | Tendon | Tendon | *Rattus norvegicus* | Wistar | No | 3 | 3 | 199 | 1,253 | 1,063 | 101 |
| TOTAL | 23 datasets (Mouse: 14, Rat: 9) | 34 tissues (Mouse: 21, Rat: 18) | 14 organs (Mouse: 12, Rat: 8) | | | | 1,173 MS runs (Mouse: 973, Rat: 200) | 211 samples (Mouse: 114, Rat: 97) | | | | |

[§]Only normal/untreated samples within this dataset are reported in this study. However, results from both normal and disease samples are available in Expression Atlas

[†] Numbers after post-processing.

taken with caution since the list is subjected to inflated False Discovery Rate (FDR), due to the accumulation of false positives when analysing the datasets separately.

Some of the organs (liver, heart and brain) were represented across multiple mouse studies in the aggregated dataset. A pairwise comparison of protein abundances in these organs generally showed a good correlation in expression (heart: $R^2$ values ranged from 0.54 to 0.83; brain: $R^2$ from 0.28 to 0.72; and liver: $R^2$ from 0.59 to 0.74) (Figs A-C in S4 File).

**2.2.2. Rat proteome.** A total of 7,769 protein groups were identified across 8 different rat organs among which 3,649 (46.9%) protein groups were unique to one specific organ while 13 (0.16%) protein groups were present among all organs (see full list in S2 File). The protein groups were mapped to 7,116 genes (canonical proteins) (S3 File). The highest number of canonical proteins (6,106, 85.1%) was found in rat kidney samples. The lowest number of canonical proteins (101, 1.4%) was found in samples from tendon, as shown in Fig 3A. The largest number of canonical proteins identified in kidney is likely because of the relatively

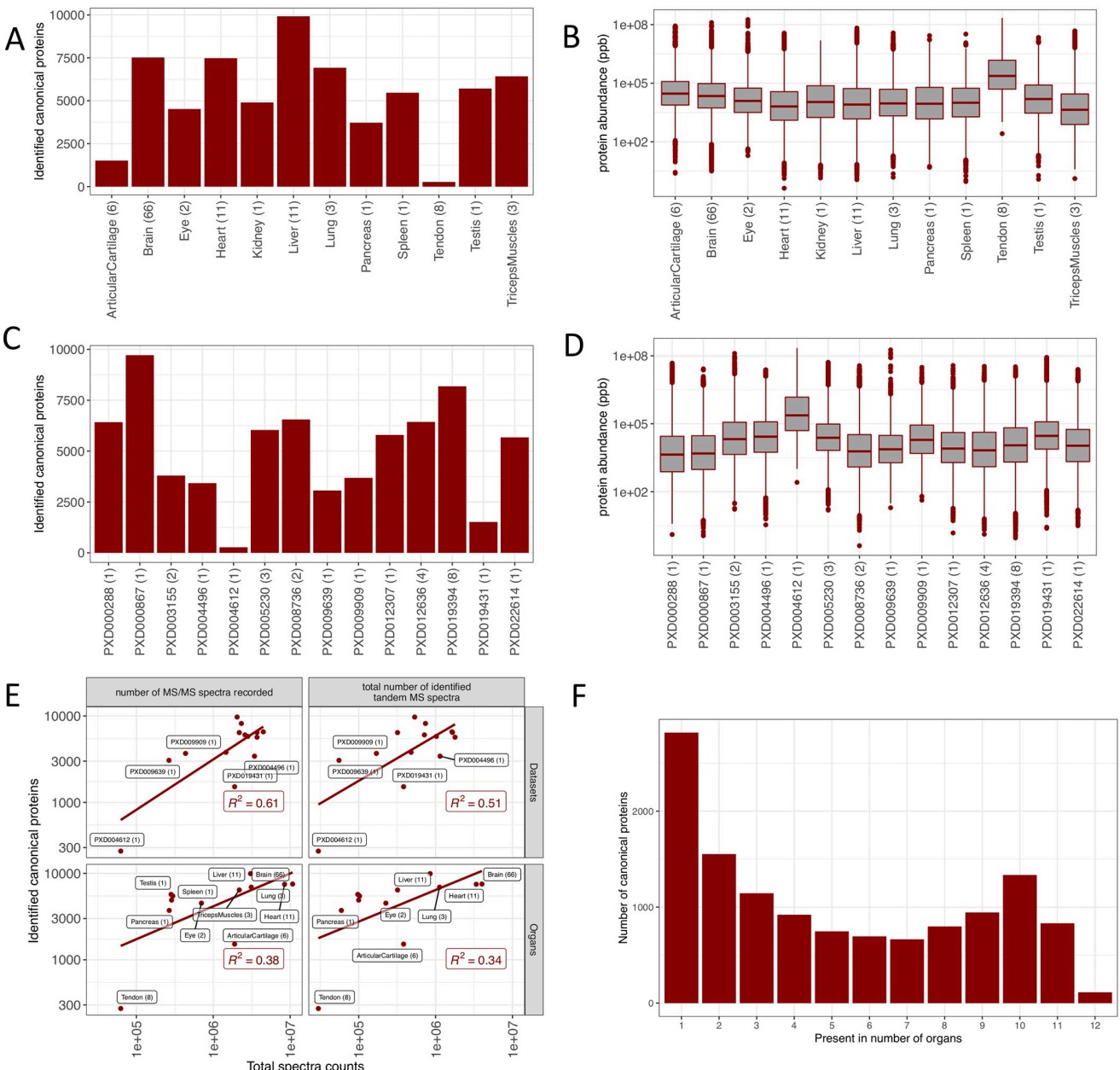

**Fig 2.** (A) Number of canonical proteins identified across different mouse organs. The number within the parenthesis indicates the number of samples. (B) Range of normalised iBAQ protein abundances across different organs. The number within the parenthesis indicates the number of samples. (C) Canonical proteins identified across different datasets. The number within the parenthesis indicate the number of unique tissues in the dataset. (D) Range of normalised iBAQ protein abundances across different datasets. The number within parenthesis indicate the number of unique tissues in the dataset. (E) Comparison of total spectral data with the number of canonical proteins identified in each dataset and organ. (F) Distribution of canonical proteins identified across organs.

large number of samples (32 samples), when compared to other organs. However, it is interesting to note that large numbers of canonical proteins were detected in liver samples, which relatively had fewer number of samples, when compared to the total number of samples in heart and spinal cord.

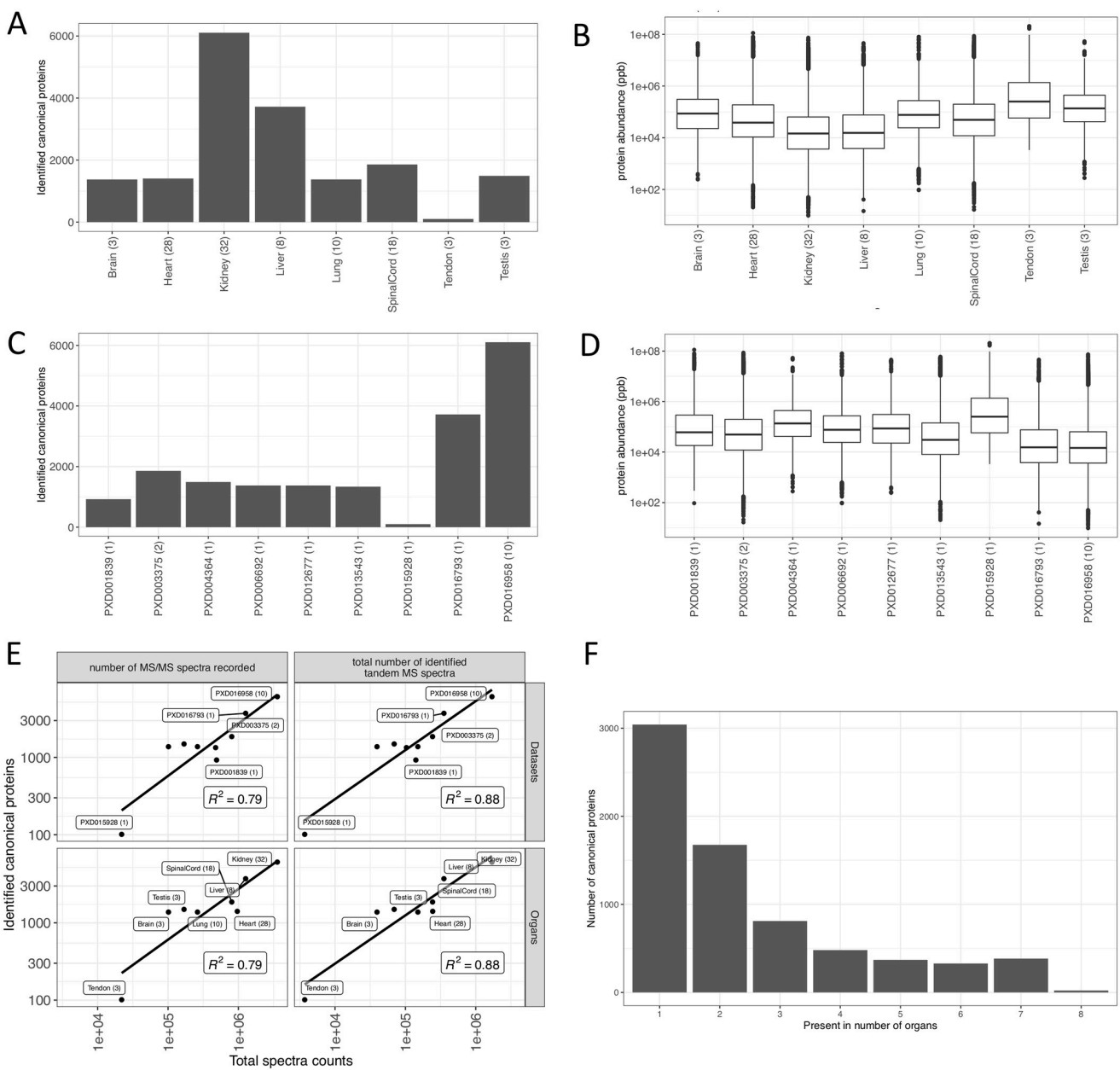

**Fig 3.** (A) Number of canonical proteins identified across different rat organs. The number within the parenthesis indicates the number of samples. (B) Range of normalised iBAQ protein abundances across different organs. The number within the parenthesis indicates the number of samples. (C) Canonical proteins identified across different datasets. The number within the parenthesis indicate the number of unique tissues in the dataset. (D) Range of normalised iBAQ protein abundances across different datasets. The number within parenthesis indicate the number of unique tissues in the dataset. (E) Comparison of total spectral data with the number of canonical proteins identified in each dataset and organ. (F) Distribution of canonical proteins identified across organs.

Datasets PXD016958 and PXD016793 consisted entirely of kidney (where fractionation was performed) and liver (no fractionation) samples, respectively, and as mentioned above had the largest number of canonical proteins identified (Fig 3C). The normalised protein abundances were similar among the various organs and datasets (Fig 3B and 3D). We also observed a linear relation between the number of canonical proteins identified and the MS spectra identified (Fig 3E). As seen in the mouse datasets, we also observed a large number of proteins uniquely

detected in one organ (Fig 3F). As highlighted above, the list of concrete canonical proteins that were detected in just one organ should be taken with caution since the list is subjected to inflated False Discovery Rate (FDR).

In the case of rat datasets, left ventricle heart samples were the only ones represented in more than one study (PXD001839 and PXD013543) in the aggregated dataset. A pairwise comparison of protein abundances of heart between these two datasets was performed, showing a strong correlation in protein expression ($R^2$ = 0.9) (Fig D in S4 File).

## 2.3. Protein abundance comparison across organs

Next, we studied how protein abundances compared across different datasets and organs. The presence of batch effects between datasets makes this type of comparisons challenging. To aid comparison of protein abundances between datasets we transformed the normalised iBAQ intensities into ranked bins as explained in 'Methods', i.e., proteins included in bin 5 are highly abundant whereas proteins in bin 1 are expressed in the lowest abundances (among the detected proteins).

**2.3.1. Mouse proteome.** We found that 1,086 (8.6%) proteins were found with their highest level of expression in at least 3 organs, with a median bin value greater than 4 (S3 File). On the other end of the scale, 138 (1.1%) canonical proteins were found with their lowest expression in at least 3 organs, with a median bin value of less than 2. The bin transformed abundances in all organs are provided in S3 File.

To compare protein expression across all organs, we calculated pairwise Pearson correlation coefficients across 117 samples (Fig 4A). We observed some correlation in protein expression within brain (median $R^2$ = 0.31) and a higher one in heart (median $R^2$ = 0.67) samples. We performed Principal Component Analysis (PCA) on all samples from mouse datasets for testing the effectiveness of the bin transformation method in reducing batch effects. Fig 4B shows the clustering of samples from various organs of mouse. We observed samples from the same organ generally clustered together. For example, we observed that brain samples all clustered together in one group, even though they come from different datasets, indicating decent removal of batch effects (Fig 4C). However, we also observed that samples from other organs such as liver did not cluster according to their organ types but clustered together within the dataset they were part of, indicating some residual batch effects, which are hard to remove completely.

In addition, we compared the protein abundances generated in this study with the data available in the resource PaxDB generated using spectral counting across different mouse organs. We observed generally a strong correlation of protein abundances calculated using iBAQ from this study (fraction of total (FOT) normalised ppb) and spectral counting methods (Fig E in S4 File). However, the expression of low abundant proteins seemed to be underestimated in PaxDB when compared with our results, as shown by a S-shaped curve in the scatterplot in organs such as brain, heart, liver and lung. The 'dynamic exclusion' [41] setting used by modern mass spectrometers prevents the instrument from fragmenting abundant peptides multiple times when they are repeatedly observed in scans nearby in time. This has the effect that spectral counting approaches will limit the dynamic range observed, as high abundant proteins will be under sampled. This is a limitation when using spectral counting methods, and these days spectral counting is not commonly used as a truly quantitative data type in proteomics.

**2.3.2. Rat proteome.** Next, we studied the distribution of protein abundances across organs in rat. On one hand, 311 (4.3%) proteins were found with their highest expression in at least 3 organs with a median bin value greater than 4. On the other hand, 27 (0.37%) canonical

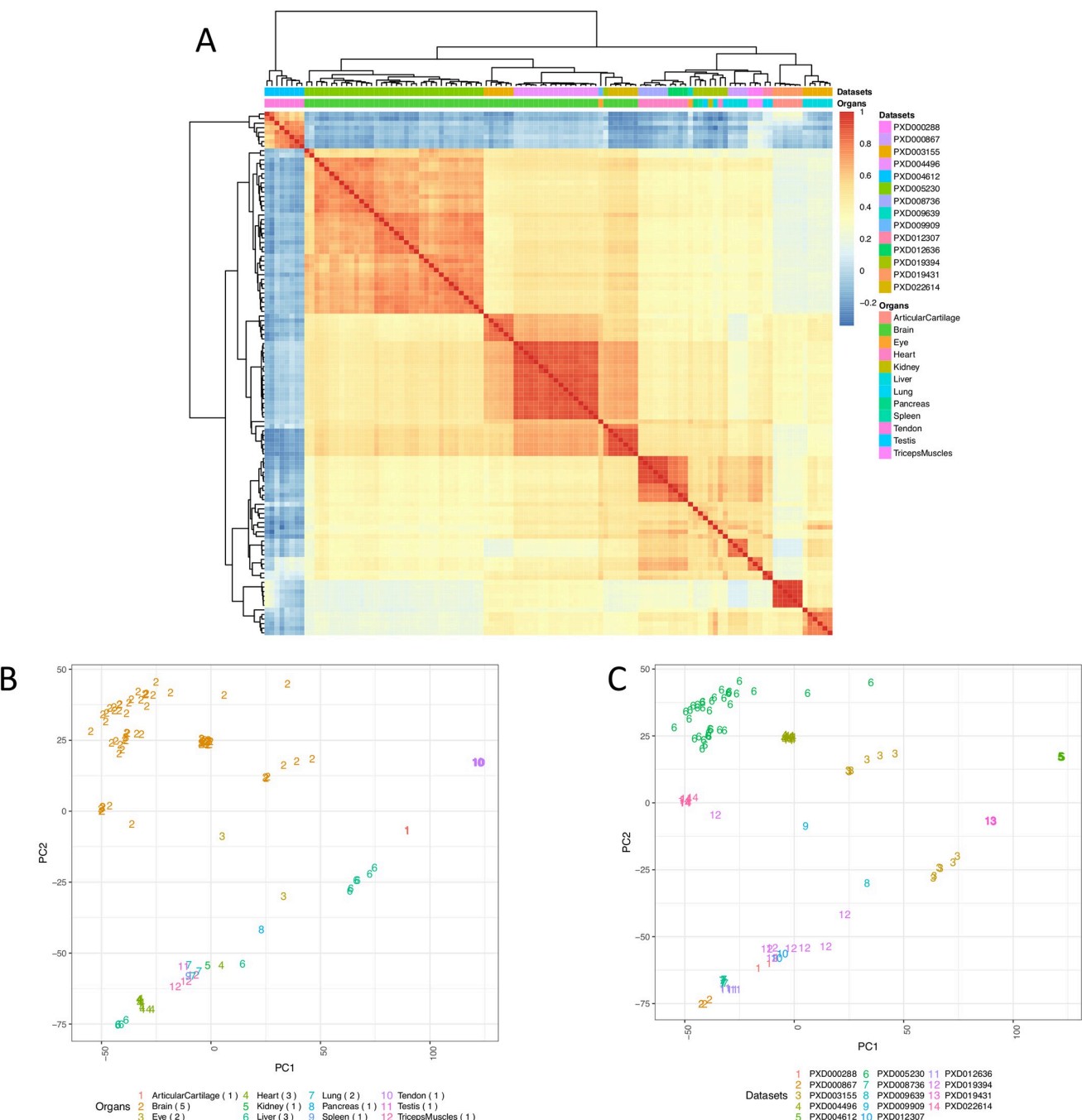

**Fig 4.** (A) Heatmap of pairwise Pearson correlation coefficients across all mouse samples. The colour represents the correlation coefficient and was calculated using the bin transformed iBAQ values. The samples were hierarchically clustered on columns and rows using Euclidean distances. (B) PCA of all samples, using the binned protein abundances as input, coloured by the organ types. (C) PCA of all samples coloured by their respective dataset identifiers. The numbers in parenthesis indicate the number of datasets for each organ. Binned values of canonical proteins quantified in at least 50% of the samples were used to perform the PCA.

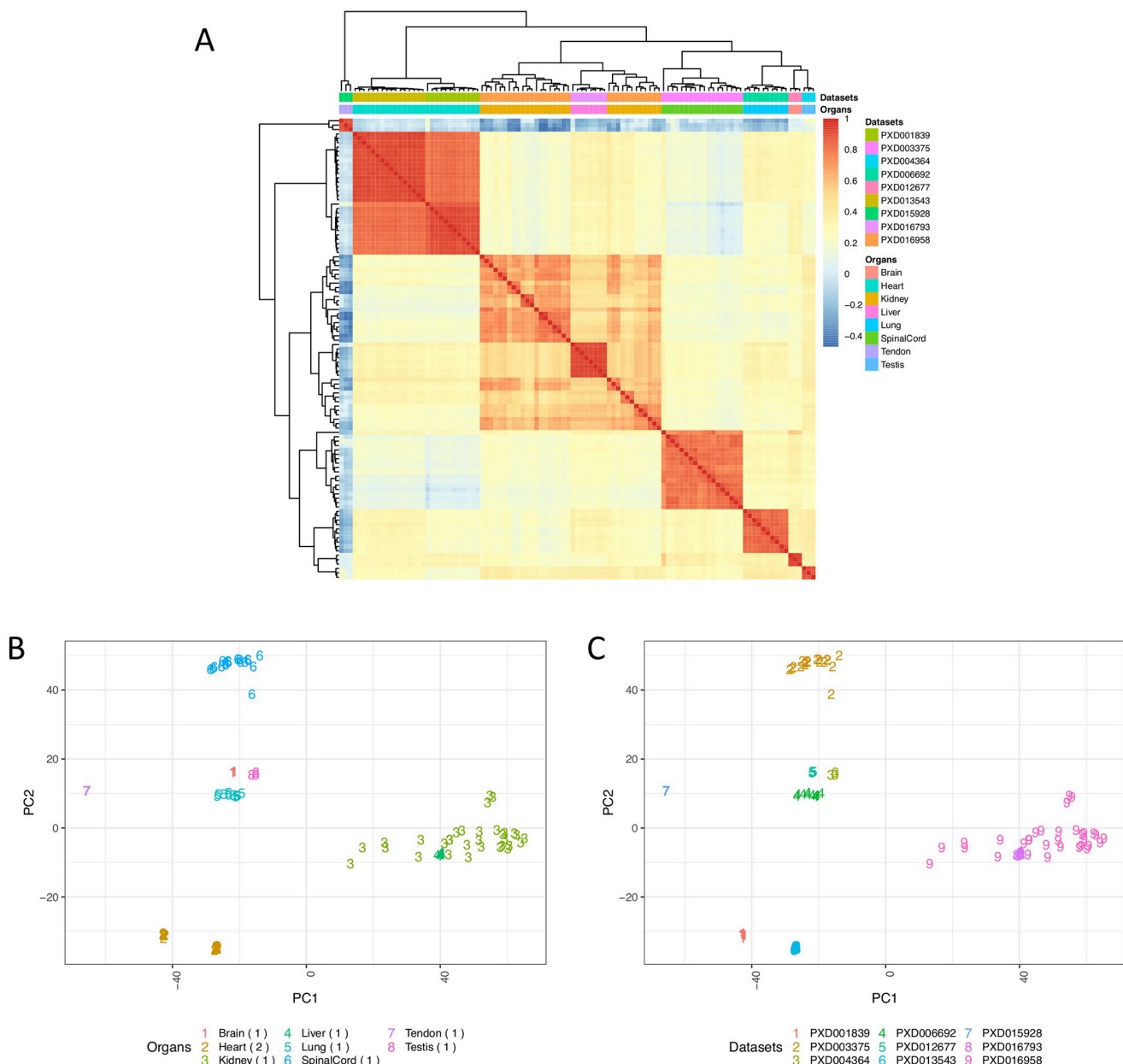

**Fig 5.** (A) Heatmap of pairwise Pearson correlation coefficients across all rat samples. The colour represents the correlation coefficient and was calculated using the bin transformed iBAQ values. The samples were hierarchically clustered on columns and rows using Euclidean distances. (B) PCA of all samples coloured by the organ types. (C) PCA of all samples coloured by their respective dataset identifiers. The numbers in parenthesis indicate the number of datasets for each organ. Binned values of canonical proteins quantified in at least 50% of the samples were used to perform the PCA.

proteins were found with their lowest expression in at least 3 organs, with a median bin value of less than 2. The bin transformed abundances in all organs are provided in S3 File.

Overall, the samples from rat datasets showed a better correlation in protein expression (Fig 5A) than in the case of mouse. We observed generally a strong correlation of protein expression within samples from liver (median Pearson's correlation $R^2 = 0.85$), lung (median $R^2 = 0.71$), spinal cord (median $R^2 = 0.65$), heart (median $R^2 = 0.71$) and brain (median $R^2 = 0.86$). We also observed the clustering in the PCA of samples coming from the same organ (Fig

5B). Kidney, lung, spinal cord and heart samples all clustered together according to their organ type. Fig 5C shows the samples based on the dataset they were part of. However, most organ samples were part of individual datasets except in the case of samples from heart, which came from two datasets (PXD001839 and PXD013543). Fig 5C shows that the heart samples clustered into two nearby groups (bottom left two clusters on Fig 5B and 5C), wherein each cluster included samples from a different dataset, indicating the presence of small batch effects.

## 2.4. The organ elevated proteome and the over-representative biological processes

Based on their expression, canonical proteins were classified into three different groups based on their organ specificity: "mixed", "group-enriched" and "organ-enriched" (see S5 File). We considered over-expressed canonical proteins in each organ as those which were in "group-enriched" and "organ-enriched". The analysis showed that on average, 20.8% and 26.0% of the total elevated canonical proteins were organ group-specific in mouse and rat, respectively (Fig 6). In addition, 4.3% and 14.2% were unique organ-enriched in mouse and rat, respectively. The highest ratio of organ-enriched in mouse was found in liver (13.6%), whereas in rat, it was found in kidney (39.8%).

We then performed a gene ontology (GO) enrichment analysis of those proteins that were 'organ-enriched' and 'group-enriched' using GO terms associated with biological processes. We found 1,036 GO terms to be statistically significant in all organs, as seen in S6 File. The most significant GO terms for each organ are shown in Table 2.

## 2.5. Protein abundances across orthologs in three species

In a previous study, we analysed 25 label-free proteomics datasets from healthy human samples to assess baseline protein abundances in 14 organs following the same analytical methodology [14]. We compared the expression of canonical proteins identified in all three species (rat, mouse and human). Overall, 13,248 detected human genes (corresponding to the canonical proteins) were compared with 12,570 genes detected in mouse and 7,116 genes detected in rat. The number of orthologous mappings (i.e., "one-to-one" mappings, see 'Methods') between rat, mouse and human genes are listed in Table 3. We only considered one-to-one mapped orthologues for the comparison of protein abundances.

Among human and mouse orthologues we observed relatively high levels of correlation of protein abundances in brain ($R^2 = 0.61$), heart ($R^2 = 0.65$) and liver ($R^2 = 0.56$) (Fig 7A).

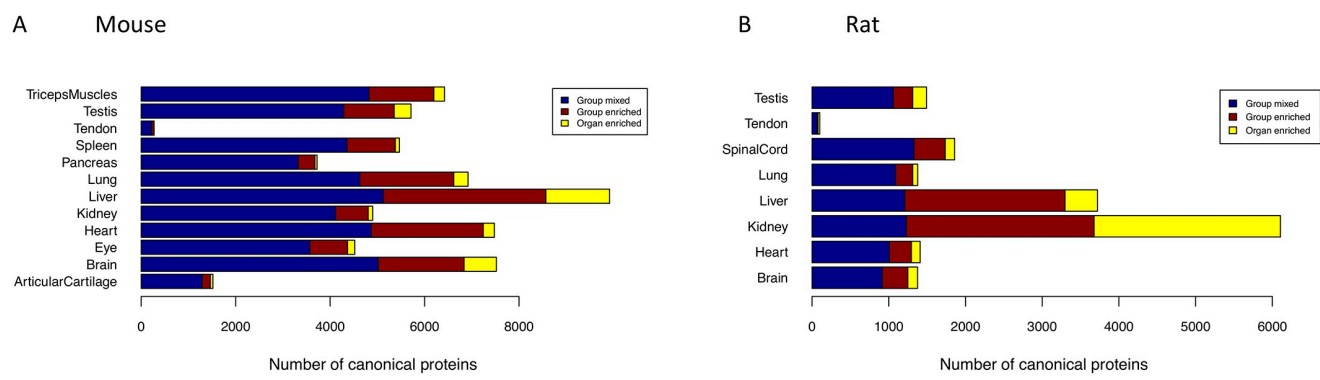

**Fig 6. Organ specificity of canonical proteins in (A) mouse and (B) rat.**

**Table 2. Analysis of the top three GO terms for each organ in mouse and rat using the elevated organ-specific and group-specific canonical proteins as described in the 'Methods' section.**

| Organ | Species | GO ID | Description | adjusted p-value |
|---|---|---|---|---|
| Articular cartilage | *Mus musculus* | GO:0030198 | Extracellular matrix organization | $8.94^*10^{-38}$ |
| | | GO:0043062 | Extracellular structure organization | $8.94^*10^{-38}$ |
| | | GO:0045229 | External encapsulating structure organization | $8.94^*10^{-38}$ |
| Brain | *Mus musculus* | GO:0050804 | Modulation of chemical synaptic transmission | $7.03^*10^{-65}$ |
| | | GO:0099177 | Regulation of trans-synaptic signalling | $7.03^*10^{-65}$ |
| | | GO:0050808 | Synapse organization | $1.41^*10^{-48}$ |
| Heart | *Mus musculus* | GO:0060047 | Heart contraction | $7.10^*10^{-11}$ |
| | | GO:0008016 | Regulation of heart contraction | $4.43^*10^{-10}$ |
| | | GO:0060537 | Muscle tissue development | $6.16^*10^{-10}$ |
| Kidney | *Mus musculus* | GO:0015711 | Organic anion transport | $4.59^*10^{-19}$ |
| | | GO:0044282 | Small molecule catabolic process | $4.91^*10^{-15}$ |
| | | GO:0016054 | Organic acid catabolic process | $6.25^*10^{-15}$ |
| Eye | *Mus musculus* | GO:0007601 | Visual perception | $7.54^*10^{-50}$ |
| | | GO:0001654 | Eye development | $5.31^*10^{-31}$ |
| | | GO:0099504 | Synaptic vesicle cycle | $8.36^*10^{-18}$ |
| Liver | *Mus musculus* | GO:0016569 | Covalent chromatin modification | $6.26^*10^{-10}$ |
| | | GO:0016570 | Histone modification | $1.71^*10^{-08}$ |
| | | GO:0019369 | Arachidonic acid metabolic process | $1.71^*10^{-08}$ |
| Lung | *Mus musculus* | GO:0120031 | Plasma membrane bounded cell projection assembly | $3.61^*10^{-14}$ |
| | | GO:0030031 | Cell projection assembly | $3.61^*10^{-14}$ |
| | | GO:0044782 | Cilium organization | $9.83^*10^{-14}$ |
| Pancreas | *Mus musculus* | GO:0007586 | Digestion | 0.005 |
| | | GO:0032328 | Alanine transport | 0.018 |
| Spleen | *Mus musculus* | GO:0046649 | Lymphocyte activation | $4.12^*10^{-22}$ |
| | | GO:0050776 | Regulation of immune response | $2.00^*10^{-20}$ |
| | | GO:0045087 | Innate immune response | $2.23^*10^{-20}$ |
| Tendon | *Mus musculus* | GO:0003012 | Muscle system process | $1.46^*10^{-25}$ |
| | | GO:0050879 | Multicellular organismal movement | $3.14^*10^{-19}$ |
| | | GO:0050881 | Musculoskeletal movement | $1.46^*10^{-25}$ |
| Testis | *Mus musculus* | GO:0048232 | Male gamete generation | $8.75^*10^{-49}$ |
| | | GO:0003341 | Cilium movement | $3.04^*10^{-38}$ |
| | | GO:0044782 | Cilium organization | $6.78^*10^{-37}$ |
| Triceps muscles | *Mus musculus* | GO:0061061 | Muscle structure development | $1.56^*10^{-14}$ |
| | | GO:0055002 | Striated muscle cell development | $2.41^*10^{-14}$ |
| | | GO:0003009 | Skeletal muscle contraction | $3.53^*10^{-14}$ |
| Brain | *Rattus norvegicus* | GO:0099537 | Trans-synaptic signalling | $1.79^*10^{-60}$ |
| | | GO:0007268 | Chemical synaptic transmission | $1.79^*10^{-60}$ |
| | | GO:0098916 | Anterograde trans-synaptic signalling | $1.79^*10^{-60}$ |
| Heart | *Rattus norvegicus* | GO:0061061 | Muscle structure development | $2.94^*10^{-17}$ |
| | | GO:0003012 | Muscle system process | $6.30^*10^{-16}$ |
| | | GO:0055001 | Muscle cell development | $4.00^*10^{-15}$ |
| Kidney | *Rattus norvegicus* | GO:0006396 | RNA processing | $6.19^*10^{-13}$ |
| | | GO:0045944 | positive regulation of transcription by RNA polymerase II | $7.29^*10^{-06}$ |
| | | GO:0006260 | DNA replication | $1.74^*10^{-05}$ |

*(Continued)*

**Table 2.** (Continued)

| Organ | Species | GO ID | Description | adjusted p-value |
|---|---|---|---|---|
| Liver | *Rattus norvegicus* | GO:0008202 | Steroid metabolic process | $2.74^*10^{-10}$ |
| | | GO:0016054 | Organic acid catabolic process | $1.61^*10^{-09}$ |
| | | GO:0032787 | Monocarboxylic acid metabolic process | $1.64^*10^{-09}$ |
| Lung | *Rattus norvegicus* | GO:0031589 | Cell-substrate adhesion | $7.62^*10^{-08}$ |
| | | GO:0009617 | Response to bacterium | $7.62^*10^{-08}$ |
| | | GO:0030036 | Actin cytoskeleton organization | $1.40^*10^{-07}$ |
| Spinal cord | *Rattus norvegicus* | GO:0061564 | Axon development | $4.26^*10^{-18}$ |
| | | GO:0099537 | Trans-synaptic signalling | $5.93^*10^{-16}$ |
| | | GO:0007268 | Chemical synaptic transmission | $5.93^*10^{-16}$ |
| Tendon | *Rattus norvegicus* | GO:0030199 | Collagen fibril organization | $1.23^*10^{-13}$ |
| | | GO:0061448 | Connective tissue development | $2.31^*10^{-09}$ |
| | | GO:0001501 | Skeletal system development | $3.39^*10^{-09}$ |
| Testis | *Rattus norvegicus* | GO:0019953 | Sexual reproduction | $3.98^*10^{-24}$ |
| | | GO:0051704 | Multi-organism process | $1.61^*10^{-18}$ |
| | | GO:0007018 | Microtubule-based movement | $4.00^*10^{-12}$ |

**Table 3. Homologs identified in mouse and rat datasets when compared with the background list of genes (corresponding to canonical proteins) identified in human datasets (Supplementary File 2 in [14]).**

| Species | Identified genes | Orthologs of human genes identified in [14] | Percentage of genes with different mapping against identified human genes | | | | |
|---|---|---|---|---|---|---|---|
| | | | one-to-one | one-to-many | many-to-many | many-to-one | not mapped |
| *Mus musculus* | 12,570 | 10,601 | 80.4% | 1.9% | 0.56% | 1.46% | 15.7% |
| *Rattus norvegicus* | 7,116 | 6,058 | 82.0% | 2.2% | 0.70% | 0.25% | 14.9% |

Human and rat orthologs showed also relatively high levels of correlation in brain ($R^2 = 0.62$), kidney ($R^2 = 0.53$) and liver ($R^2 = 0.56$), but almost no correlation in lung ($R^2 = 0.12$) and testis ($R^2 = 0.18$) (Fig 7B). Between mouse and rat orthologs, the correlation of protein abundances was higher in liver ($R^2 = 0.65$), kidney ($R^2 = 0.54$) and brain ($R^2 = 0.57$) samples, when compared to the samples coming from the rest of the organs (Fig 7C). Fig 7D shows an illustration of some example comparisons of individual orthologs using binned protein abundances.

For the same corresponding subsets, we also investigated the correlation of protein expression between various organs within each organism. We observed that in general the correlation of protein expression was slightly lower between organs within the same species, when compared to a higher correlation, which was observed among orthologs (Figs F-H in S4 File). The found lower correlation of protein expression between different organs was more apparent in mouse and rat.

Among the orthologs expressed in all organs in all three species, 747 (12.3%) orthologs were detected with a median bin expression value of more than 4, i.e., proteins that appear to have conserved high expression in all organs and all tissues. Additionally, 13 (0.2%) orthologs were found with a median bin expression value less than 2 in all organs, although, it is harder to detect consistently proteins with low abundances across all organs. A full list of the binned protein abundances of orthologs is available in S7 File. The illustration of all binned protein abundances across the three species is shown in S8 File.

Since each sample contains potentially thousands of protein values this creates a high level of dimensionality within the data. To reduce this, we used the non-linear dimension reduction

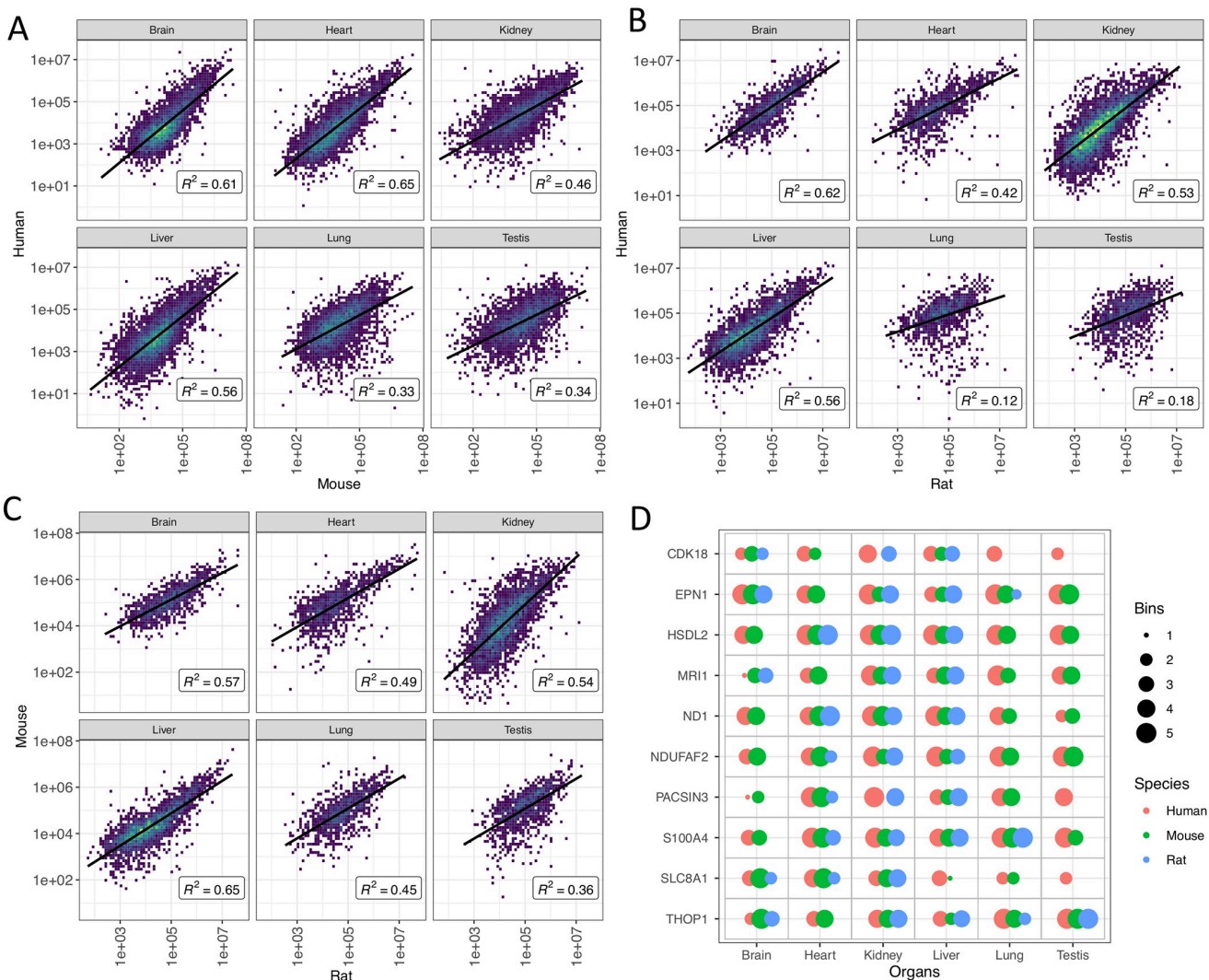

**Fig 7. Comparison of protein abundances (in ppb) between one-to-one mapped orthologs of mouse, rat and human in various organs.** (A) Pairwise correlation using normalised protein abundances of human and mouse orthologues. (B) Human and rat orthologs. (C) Mouse and rat orthologs. (D) As an example, the comparisons of binned protein expression of ten randomly sampled orthologs are shown. Data corresponding to all cases (as reported in panel D) are available in S7 File and the corresponding illustration of binned values is available in S8 File. Orthologs in (D) are shown using their human gene symbol.

algorithm, Uniform Manifold Approximation and Projection (UMAP) (see Section 4.7 in the 'Methods' section). The UMAP algorithm enables the reduction of multidimensional data to a two-dimensional space upon which the relationship between each sample can be visualised. Specifically, it enables the visualisation of the relationships of proteins across individual samples and organs. Should multiple samples be positioned near to each other, it allows for us to predict that these samples shared similar properties (in this case, similar protein abundance values). Consequently, by overlaying samples from various species UMAP representations can be used to visualise the relationship of various orthologs across similar organs.

Using the UMAP algorithm, we were able to visualise the relationships between individual organs regardless of the involved species (human, mouse, rat) and to identify similar genes (corresponding to canonical proteins) within those organs. The overall view of all samples labelled by their respective organ is shown as Fig 8A. We chose to use the biological system as

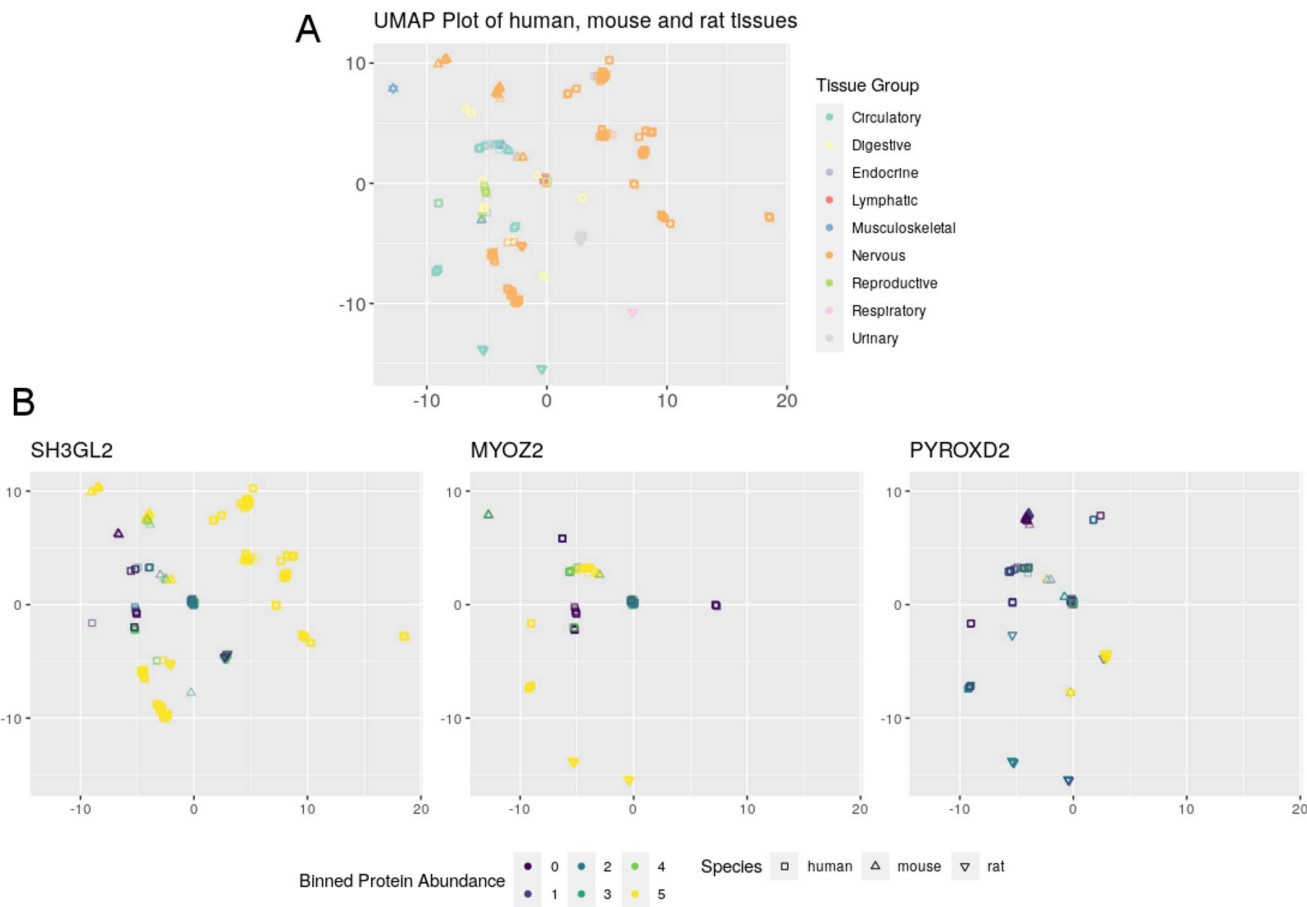

**Fig 8. Visualisations generated using the UMAP algorithm to show the relationships between human, mouse, and rat samples.** (**A**) Shows the relationship of all samples, particularly showing strong relationship between biological systems. (**B**) Shows the protein abundancy of 3 example gene orthologs (SH3GL2, MYOZ2 and PYROXD2), within each sample. Human baseline protein expression data was generated in [14].

the basis for the colouring scheme for each sample to reduce the overall complexity of the visualisation, due the high number of organs included. By using this labelling scheme, we could see that the clustering of each sample was deterministic. Each sample was positioned within a clear region for the corresponding organs, despite the original layout being unaware of this information. This indicates that not only do the samples within those organs share common protein abundance values, but furthermore, that samples that come from the same organs share similar protein expression (as three species are present).

Furthermore, in Fig 8B we show the representation of binned protein abundance values for three example genes (SH3GL2, MYOZ2 and PYROXD2), providing information on the abundance of them across different biological systems. These visualisations use the same layout than within Fig 8A. In the example of SH3GL2, it can be seen that Fig 8B shows multiple values that have been scored as bin 5. By referring to Fig 8A, we can see that those points corresponding to highly abundant proteins, come from samples from the nervous system (in all three species). Furthermore, using the same method, it can be seen that MYOZ2 is highly abundant in the circulatory system, and that PYROXD2 is highly abundant in the urinary system. The UMAP coordinates and our binned protein abundance data that is used in these plots to allow for the generation of similar visualisations are provided in S9 File.

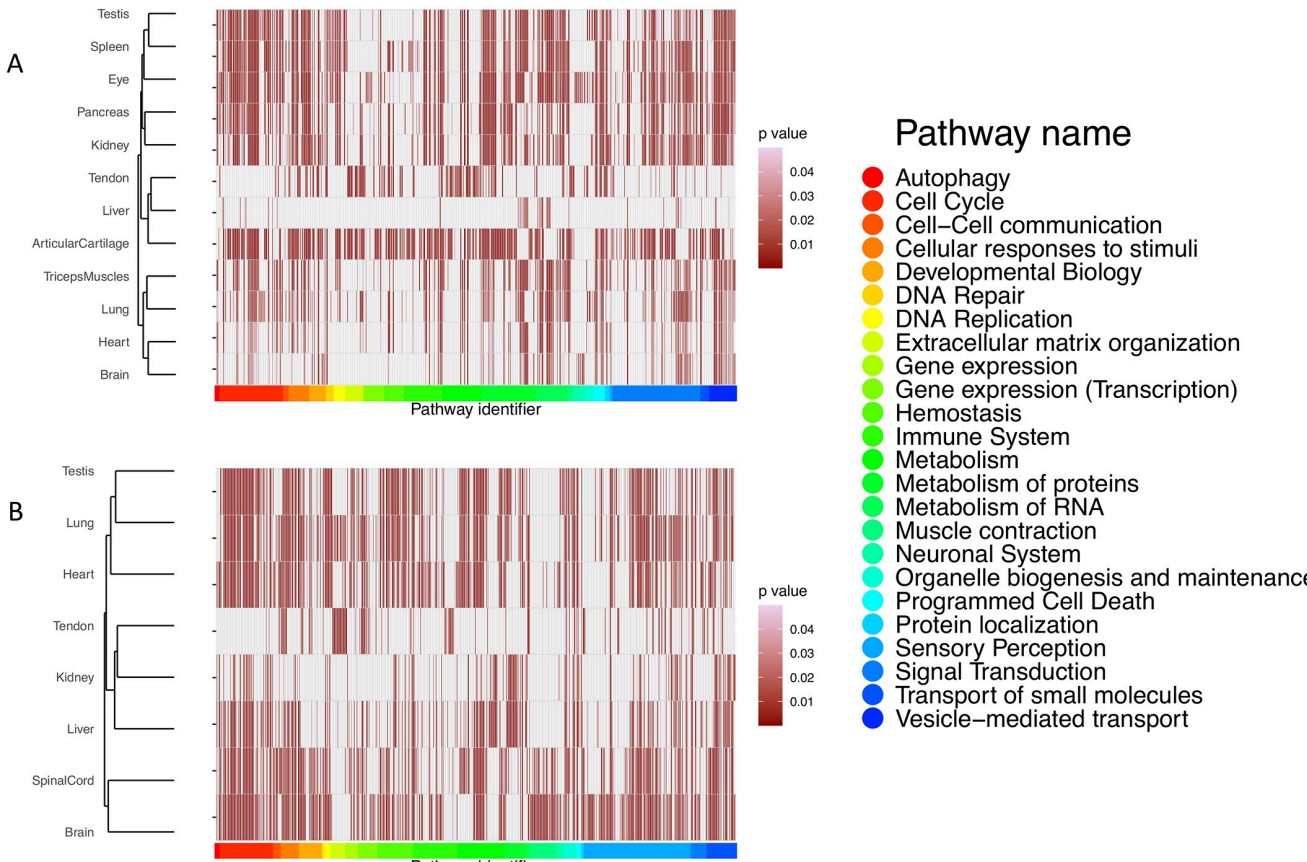

**Fig 9. Pathway analysis performed using the canonical proteins, showing the statistically significant representative pathways (p-value < 0.05) in (A) mouse and (B) rat organs.**

## 2.6. Pathway enrichment analysis

Based on the ortholog protein expression analysis described above, we mapped canonical proteins from mouse and rat to the corresponding ortholog human proteins, which were subsequently subjected to pathway-enrichment analysis using Reactome (Fig 9). After filtering out the disease and statistically insignificant pathways, there were 2,990 pathways found in all the organs of mouse and 2,162 pathways in all the organs of rat. In mouse samples, the largest number of pathways (367) were found in articular cartilage, and the lowest number of pathways was found in liver (44). We also observed that Neuronal System-related pathways were predominantly present in the brain and eye, which is consistent with expectations. In rat samples, brain included the largest number of pathways (387), while the lowest number of pathways was found in tendon, with 117.

## 3. Discussion

We have previously reported two meta-analysis studies involving the reanalysis and integration in Expression Atlas of public quantitative datasets coming from cell lines and human tumour samples [13], and from human baseline tissues [14], respectively. In this study, we reanalysed mouse and rat baseline proteomics datasets representing protein expression across 34 healthy tissues and 14 organs. We have used the same methodology as in the study involving

baseline human tissues, which enabled a comparison of protein expression levels across the three species. Our main overall aim was to provide a system-wide baseline protein expression catalogue across various tissues and organs of mouse and rat and to offer a reference for future related studies.

We analysed each dataset separately using the same software (MaxQuant) and the same search protein sequence database. The disadvantage of this approach is that the FDR statistical thresholds are applied at a dataset level and not to all datasets together as a whole. However, as reported before [14], using a dataset per dataset analysis approach is in our view the only sustainable manner to reanalyse and integrate quantitative proteomics datasets, at least at present. The disadvantage of this approach is that the FDR statistical threshold are applied at a dataset level and not to all datasets together as a whole, with the potential accumulation of false positives across datasets. However, it is important to highlight that the number of commonly detected false positives is reduced in parallel with the increase in the number of common datasets where a given protein is detected. As also reported in previous studies, one of the major bottlenecks was the curation of dataset metadata, consisting of mapping files to samples and biological conditions. Very recently, the MAGE-TAB-Proteomics format has been developed and formalised to enable the reporting of the experimental design in proteomics experience, including the relationship between samples and raw files, which is recorded in the SDRF-Proteomics section of the file [42]. Submission of the SDRF-Proteomics files to PRIDE is now supported. The more well-annotated datasets in the public domain, the easier these data reuse activities will become.

The generated baseline protein expression data can be used with different purposes such as the generation of protein co-expression networks and/or the inference of protein complexes. For the latter application, expression data can be alone or for potentially refining predictions obtained using different methods such as the recently developed AlphaFold-based protein complexes predictions [43]. Mouse and rat are widely used species in the context of drug discovery, the latter especially, to undertake regulatory pre-clinical safety studies. Therefore, it is important to know quantitative protein expression distribution in these species in different tissues [44] to assist in the selection of species for such studies and also for the interpretation of the final results.

In addition to the analyses reported, it would have also been possible to perform correlation studies between gene and protein expression information. However, we did not find any relevant public datasets in the context of this manuscript where the same samples were analysed by both techniques, which is the optimal way to perform these studies. Future directions in analogous studies will involve: (i) additional baseline protein expression studies of other species, including other model organisms or other species of economic importance; (ii) the inclusion of differential proteomics datasets (e.g. using TMT and/or iTRAQ); and (iii) include relevant proteomics expression data coming from the reanalysis of Data Independent Acquisition (DIA) datasets [45].

As mentioned above, we performed a comparative analysis of baseline protein expression across human, mouse and rat. It was possible to perform this analysis for six common organs (brain, heart, kidney, liver, lung and testis). Ortholog expression across species is useful to infer protein function across experimentally studied proteins. This is particularly useful as evolutionarily closely related species are likely to conserve protein function. We could not find in the literature an analogous comparative study performed at the protein level. However, expression from closely related orthologs across tissues or organs has been compared at the transcriptomics level, providing a complete picture of gene expression. In this context, many studies have compared gene-expression in mouse, rat and human orthologues and found that orthologues had generally a highly correlated expression tissue distribution profile in baseline

conditions [46–50]. Gene expression levels among orthologs were found to be highly similar in muscle and heart tissues, liver and nervous system and less similar in epithelial cells, reproductive systems, bone and endocrine organs [48]. Studies have also shown that variability of gene expression between homologous tissues/organs in closely related species can be lower than the variability between unrelated tissues within the same organism [46,47], in agreement with the results reported here at the protein level. Additionally, we showed an initial analysis of protein expression of orthologs across the three species using UMAP.

In conclusion we here present a meta-analysis study of public mouse and rat baseline proteomics datasets from PRIDE. We demonstrate its feasibility, perform a comparative analysis across the three species and show the main current challenges. Finally, the data is made available *via* Expression Atlas. Whereas there are several analogous studies performed at the gene expression level for mouse and rat tissues, to the best of our knowledge this is the first of this kind at protein expression level.

## 4. Materials and methods

### 4.1. Datasets

As of May 2021, there were 2,060 mouse (*Mus musculus*) and 339 rat (*Rattus norvegicus*) MS proteomics datasets publicly available in the PRIDE database (https://www.ebi.ac.uk/pride/). Datasets were manually selected based on the selection criteria described previously [14]. Briefly, we selected datasets where baseline expression experiments were performed on (i) label-free samples from tissues not enriched for post-translational modifications; (ii) Thermo Fisher Scientific instruments such as LTQ Orbitrap, LTQ Orbitrap Elite, LTQ Orbitrap Velos, LTQ Orbitrap XL ETD, LTQ-Orbitrap XL ETD, Orbitrap Fusion and Q-Exactive, since they represent a large proportion of datasets in PRIDE and to avoid heterogeneity introduced by data from other vendor instruments; (iii) had suitable sample metadata available in the original publication or it was possible to obtain it by contacting the authors; and (iv) our previous experience in the team of some datasets deposited in PRIDE, which were discarded because they were not considered to be useful. Overall, 14 mouse and 9 rat datasets were selected from all mouse and rat datasets for further analysis. Table 1 lists the selected datasets. The 23 datasets contained a total of 211 samples from 34 different tissues across 14 organs (meaning groups of related tissues, more details below), comprising 9 different mouse and 3 rat strains, respectively.

The sample and experimental metadata were manually curated using the information provided in the respective publications or by contacting the original authors/submitters. Annotare [51] was used for annotating the metadata and stored using the Investigation Description Format (IDF) and Sample-Data Relationship Format (SDRF) file formats [42], which are required for integration in Expression Atlas. An overview of the experimental design including experimental factors, protocols, publication information and contact information are present in the IDF file, and the SDRF includes sample metadata describing the relationship between the various sample characteristics and the data files contained in the dataset.

### 4.2. Proteomics raw data processing

All datasets were analysed with MaxQuant (version 1.6.3.4) [52,53] on a Linux high-performance computing cluster for peptide/protein identification and protein quantification. Input parameters for each dataset, such as $MS^1$ and $MS^2$ tolerances, digesting enzymes, fixed and variable modifications, were set as described in their respective publications, with two missed cleavage sites. The FDR at the PSM (peptide spectrum match) and protein levels were set to 1%. The MaxQuant parameters were otherwise set to default values: the maximum number of

modifications per peptide was 5, the minimum peptide length was 7, the maximum peptide mass was set to 4,600 Da, and for the matches between runs the minimum match time window was set to 0.7 seconds and the minimum retention time alignment window was set to 20 seconds. The MaxQuant parameter files are available for downloading from Expression Atlas. The *Mus musculus* UniProt Reference proteome release-2021_04 (including isoforms, 63,656 sequences) and *Rattus norvegicus* UniProt Reference proteome release-2021_04 (including isoforms, 31,562 sequences) were used as the target sequence databases for mouse and rat datasets, respectively. The built-in contaminant database within MaxQuant was used and a decoy database was generated by MaxQuant by reversing the input database sequences after the respective enzymatic digestion. The datasets were run separately in multi-threaded mode.

### 4.3. Post-processing

The post-processing of results from MaxQuant is explained in detail in [14]. In brief, the protein groups labelled as potential contaminants, decoys and those with fewer than 2 PSMs were removed. Protein intensities in each sample were normalised by scaling the iBAQ intensity values to the total amount of signal in each MS run and converted to parts per billion (ppb).

$$ppb\_iBAQ_i = \left( {iBAQ_i} \Big/ {\sum_{i=1}^{n} iBAQ_i} \right) x\ 1,000,000,000$$

The 'majority protein identifiers' within each protein group were mapped to their Ensembl gene identifiers/annotations using the Bioconductor package 'mygene'. For downstream analysis only protein groups whose isoforms mapped to a single unique Ensembl gene ID were considered. Protein groups that mapped to more than one Ensembl gene ID are provided in S1 File. The protein intensity values from different protein groups with the same Ensembl gene ID were aggregated as median values. The parent genes to which the different protein groups were mapped to are equivalent to 'canonical proteins' in UniProt (https://www.uniprot.org/help/canonical_and_isoforms) and therefore the term protein abundance is used to describe the protein abundance of the canonical protein throughout the manuscript.

### 4.4. Integration into expression atlas

The calculated canonical protein abundances (mapped to genes), together with the validated SDRF files, summary files detailing the quality of post-processing and the input MaxQuant parameter files (mqpar.xml) were integrated into Expression Atlas (https://www.ebi.ac.uk/gxa/home) as proteomics baseline experiments (E-PROT identifiers are available in Table 1).

### 4.5. Protein abundance comparison across datasets

To compare protein abundances, the normalised protein abundances (in ppb) from each group of tissues in a dataset were converted into ranked bins. In this study, 'tissue' is defined as a distinct functional or structural region within an 'organ'. For example, hippocampus, cerebellum and cortex are defined as 'tissues' that are part of the brain (organ) and similarly sinus node, left atria, left ventricle, right atria, right ventricle are defined as 'tissues' in heart (organ). Protein abundances were transformed into bins by first grouping MS runs from each tissue within a dataset as a batch. The normalised protein abundances (ppb) for each MS run within a batch were sorted from lowest to highest abundance and ranked into 5 bins. Proteins whose ppb abundances are ranked in the lowest bin (bin 1) represent lowest abundance and correspondingly proteins within bin 5 are of highest abundance in their respective tissue. When merging tissues into organs, median bin values were used.

Proteins that were detected in at least 50% of the samples were selected for PCA (Principal Component Analysis) and was performed using R (The R Stats package) [54] using binned abundance values. For generating heatmaps, a Pearson correlation coefficient for all samples was calculated on pairwise complete observations of bin transformed values. Missing values were marked as NA (not available). For each organ a median $R^2$ was calculated from all pairwise $R^2$ values of their respective samples. Samples were hierarchically clustered on columns and rows using Euclidean distances. To compare the correlation in protein expression of shared organs between datasets, the FOT normalised protein abundances (ppb) were aggregated by calculating the median over samples. The regression line was computed using the 'linear model' (lm) method in R.

## 4.6. Comparison of protein abundances using iBAQ and spectral counting data available in PaxDB

To compare protein abundances generated from iBAQ in this study and spectral counting methods, protein abundance data from different mouse organs was obtained from PaxDB (https://www.pax-db.org/) [16]. FOT normalised iBAQ abundances, as described above, were compared with the spectral counting abundances for the matching mouse organs. Organs from mouse labelled as 'integrated' in PaxDB were selected. It was not possible to perform this comparison for rat organs since data in PaxDB for rat are available for either the 'whole organism' or for "cell types" only. Abundances were compared across mouse adipose tissue, brain, heart, kidney, liver, lung, pancreas and spleen. The Ensembl ENSG gene ids were mapped to ENSP protein ids in PaxDB using the 'mygene' bioconductor package in R.

## 4.7. UMAP analysis

To generate the UMAP visualisations we used the binned protein abundance values generated in this study from rat and mouse, as well as the binned human protein abundance values from [14]. First, we reduced this data to only contain the orthologs found in all three species. For the purpose of only the initial visualisation layout, we filtered the data to include those proteins present in 90% of samples. Once the initial layout was generated, we then used the full protein abundance values to generate protein-specific visualisations. We use R v4.1.0 with the package 'umap' (Uniform Manifold Approximation and Projection in R) [55] v0.2.7.0 to generate the UMAP visualisations.

## 4.8. Organ-specific expression profile analysis

For comparison across organs, the tissues were aggregated into organs and their median bin values were considered. As described previously [14] the classification scheme done by Uhlén *et al.* [17] was modified to classify the proteins into one of the three categories: (1) "Organ-enriched": present in one unique organ with bin values 2-fold higher than the mean bin value across all organs; (2) "Group enriched": present in at least 7 organs in mouse or in at least 4 organs in rat, with bin values 2-fold higher than the mean bin value across all organs; and (3) "Mixed": the remaining canonical proteins that are not part of the above two categories.

Enriched gene ontology (GO) terms analysis was carried out through over-representation test described previously [14], it was combined with "Organ-enriched" and "Group enriched" mapped gene lists for each organ. In addition, Reactome [56] pathway analysis was performed using mapped gene lists and running pathway-topology and over-representation analysis, as reported previously [14].

### 4.9. Comparison of protein expression across species

The g:Orth Orthology search function in the g:Profiler suite of programs [57] was used for translating gene identifiers between organisms. Since a custom list of gene identifiers could not be used as the background search set, the mouse and rat genes were first mapped against the background Ensembl database. The resulting list of mouse and rat genes mapped to human orthologs were then filtered so that they only included parent gene identifiers of the protein groups from mouse and rat organs identified in this study and the parent genes of human organs described in our previous study (Supplementary File 2 in [14]), respectively.

The orthologs were grouped into various categories denoting the resulting mapping between identifiers: "one-to-one", "one-to-many", "many-to-one", "many-to-many", and "no mappings" between gene identifiers. Only "one-to-one" mapped ortholog identifiers were used to compare protein intensities between mouse, rat and human organs. The normalised ppb protein abundances of the one-to-one mapped orthologues in 6 organs (brain, heart, kidney, liver, lung and testis), that were studied across all three organisms were used to assess the pairwise correlation of protein abundances. The linear regression was calculated using the linear fit 'lm' method in R.

## Supporting information

**S1 File. Protein groups from all datasets that are mapped to more than one Ensembl Gene ID.**
(XLSX)

**S2 File. Median protein abundances (in ppb) for each protein group across various tissue samples in each organ.**
(XLSX)

**S3 File. Median binned protein abundances across various tissue samples in each organ of mouse and rat.**
(XLSX)

**S4 File.** Supplementary figures (A) illustrating correlation of protein abundances in organs represented in different datasets. (B) Correlation of protein abundances generated using iBAQ and spectral counting methods in various mouse organs. (C) Correlation of protein expression between organs within human, mouse and rat.
(PDF)

**S5 File. Organ distribution of canonical proteins in mouse and rat.**
(XLSX)

**S6 File. Gene Ontology enrichment analysis of 'organ-enriched' and 'group-enriched' proteins.**
(XLSX)

**S7 File. Binned protein abundances of one-to-one mapped orthologs across all organs studied.**
(XLSX)

**S8 File. Figure illustrating binned protein abundances of all one-to-one mapped orthologs across six common organs in mouse, rat and human.**
(PDF)

**S9 File. UMAP co-ordinates and source data of UMAP analysis.**
(XLSX)

## Acknowledgments

First of all, we would like to thank all data submitters who made their datasets available via PRIDE and ProteomeXchange. We would also like to thank Andrew Leach and the rest of the team involved in the Open Targets "Target Safety" project, for helpful discussions.

## Author Contributions

**Conceptualization:** David García-Seisdedos, Ananth Prakash, Juan Antonio Vizcaíno.

**Data curation:** Shengbo Wang, David García-Seisdedos, Ananth Prakash, Deepti Jaiswal Kundu, Nancy George, Silvie Fexova.

**Formal analysis:** Shengbo Wang, David García-Seisdedos, Ananth Prakash.

**Funding acquisition:** Irene Papatheodorou, Andrew R. Jones, Juan Antonio Vizcaíno.

**Investigation:** David García-Seisdedos, Juan Antonio Vizcaíno.

**Project administration:** Irene Papatheodorou, Juan Antonio Vizcaíno.

**Resources:** David García-Seisdedos, Ananth Prakash.

**Software:** Shengbo Wang, David García-Seisdedos, Ananth Prakash, Andrew Collins, Andrew R. Jones.

**Supervision:** Irene Papatheodorou, Andrew R. Jones, Juan Antonio Vizcaíno.

**Validation:** Shengbo Wang, David García-Seisdedos, Ananth Prakash, Nancy George, Silvie Fexova, Pablo Moreno.

**Visualization:** David García-Seisdedos, Ananth Prakash, Nancy George, Silvie Fexova, Pablo Moreno.

**Writing – original draft:** Shengbo Wang, David García-Seisdedos, Ananth Prakash, Juan Antonio Vizcaíno.

**Writing – review & editing:** Shengbo Wang, David García-Seisdedos, Ananth Prakash, Andrew Collins, Irene Papatheodorou, Andrew R. Jones, Juan Antonio Vizcaíno.

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
