## [Decision Letter · Decision Letter 0]

27 Jan 2022

Dear Dr. Prakash,

Thank you very much for submitting your manuscript "Integrated view and comparative analysis of baseline protein expression in mouse and rat tissues" for consideration at PLOS Computational Biology.

As with all papers reviewed by the journal, your manuscript was reviewed by members of the editorial board and by several independent reviewers. In light of the reviews (below this email), we would like to invite the resubmission of a significantly-revised version that takes into account the reviewers' comments.

We cannot make any decision about publication until we have seen the revised manuscript and your response to the reviewers' comments. Your revised manuscript is also likely to be sent to reviewers for further evaluation.

Sincerely,

Zhaolei Zhang

Associate Editor

PLOS Computational Biology

Ilya Ioshikhes

Deputy Editor

PLOS Computational Biology

Reviewer's Responses to Questions

**Comments to the Authors:**

Reviewer #1: This manuscript integrated mouse and rat proteome datasets from published works and performed some correlation-based analysis across organs and species. I have the following questions regarding this manuscript, and hopes the authors could address.

1. The paper seems to be a following up work from the group’s previous paper (Jarnuczak, A.F., Najgebauer, H., Barzine, M. et al. An integrated landscape of protein expression in human cancer. Sci Data 8, 115 (2021). https://doi.org/10.1038/s41597-021-00890-2), but applies to mouse/rat proteomics data. The original work focused on human cell lines. The concepts and methodologies of the two papers are extremely similar. It would be nice if the authors could highlight the computational innovations introduced in the current paper not in the previous one.

2. To my knowledge, PaxDB (https://pax-db.org/) is a very popular database for checking protein abundance. PaxDB contains many widely studied species including mouse and rat. It also has tissue information, protein interaction information and is regularly updated for years. To me, PaxDB seems to cover all values this paper could provide to me. It would nice that the authors could highlight the advantages of this paper that PaxDB doesn’t have.

3. One issue related to DDA based bottom-up proteomics is the reproducibility. One comparison that is missing in this paper is how well the proteins correlate with each other from different datasets for the same organ. This is critical. If the common proteins sampled from the same organ from two individual studies do not well correlate with each other, then to establish a global baseline for proteome across organs and species would be not very useful. For example, is TMT labelling different from iTRAQ labelling? Is deep fractionation proteome different from non-fractionation proteome? How do the authors adjust the difference using any kind of statistical modelling?

4. The title of this paper is about baseline expression of proteins. How did the baseline is set? By just the mean of expression or with any statistical justification?

5. How was the pathway analysis performed? If it was performed like gene set enrichment analysis, of course you will identify so many pathways with significant p-values. This will not provide much values. The interesting thing to see would be if particular pathways are detected in specific organs/tissue, but not in the others.

6. I am not super on top of the current mouse/rat proteomics literature. I am not sure if any targeted/DIA proteomics work has been done in mouse or rat. It would be nice to benchmark the DDA proteome to targeted/DIA proteome, since it was argued that targeted/DIA proteome measure is more accurate than DDA proteome.

Reviewer #2: Wang et al. described the results from a comparative analysis of publicly available rat and mouse proteomics data sets generated for fourteen tissues in the baseline (healthy) state. They verified that nearly half the detected proteins were significantly over-expressed in one or two types of tissue/organ system, and certain tissues such as tendon and testis have different sets of proteins dominating the abundance distribution given the uniqueness of their physiological and biological functions. They also noted that the protein expression levels are highly correlated between orthologs across species in line with general expectation. They are aware of the potential batch effects as individual data sets profile specific target organs only, and not all tissues and organs were analyzed in one single experiment. All in all, re-analyzing >20 proteomics datasets and standardizing the identification and quantification results (e.g. binning abundance levels) is a giant undertaking, and technical aspects of the work look solid in my opinion. Having said that, I believe the manuscript could have added more informative and exciting data analysis, rather than ending with the casual analysis of ortholog correlations (Figure 7) and generic functional enrichment-based clustering of organs (Figure 8).

Major recommendations

• Expression Atlas has a large number of human proteomics data sets as well. One way to utilize the resource in the context of this paper would be to map unambiguous orthologs across the three species (as much as one can) in similar organs and tissues, and perform a projection analysis of proteins (e.g. t-SNE or UMAP) all at once. In such a visualization, for instance, serum amyloid A1 proteins should be almost uniquely synthesized in hepatocytes of the liver, and this protein from the three species should co-localize to the same proximal neighborhood in the projection plot (they can be labeled as SAA1_rat, SAA1_mouse, SAA1_human). It will be an interesting exercise to catalogue what proteins are quantitatively enriched in particular organ systems across the three species, and what proteins are not – the latter of which is no doubt the more intriguing part of the results. Describing the consistent and inconsistent findings across the species for endocrine (liver, pancreas, kidney, adrenal glands) and immune systems (spleen, if you have it) will be very useful for many investigators working on the molecular pathophysiology of a disease in specific organ system.

• I wonder if it is possible to acquire or assemble similar baseline mRNA expression data sets for matching sample types (MGI for mouse, RGD for rat, GTEx for human). This will allow you to evaluate tissue specific mRNA-protein ratio comparisons between species. While the lack of absolute quantification precludes the calculation of protein translation rates, comparison of pseudo-ratios of protein/mRNA across organ systems may turn out to be divergent across the species (or not).

Minor comments

• Unlike tissue specific mRNA expression data sets (e.g. RNA-seq), MS/MS-based proteomics analyses report identifiable, mostly soluble fraction of the proteome, which may differ by tissue types. For this reason, the current proportion-based normalization (ppb-iBAQ) may underestimate the missing fraction of the proteome in the denominator, i.e. the sum of all quantified proteins in that analysis of the tissue sample. In my humble opinion, the denominator should add a tissue-specific fudge factor to the sum, if one can estimate it. For instance, if you can find matching RNA-seq data sets, you can look at the overlap between identified proteins and the number of genes whose mRNA is expressed >1 in TPM in each tissue/organ type. This will reveal the fraction of identified and unidentified proteins, and you can add the estimate of the missing proteome abundance to the denominator. Of course this will require huge assumptions such as mRNA and protein levels are generally linearly correlated. I wonder whether this is a worthy investigation, or of interest to the authors. If you believe that the current normalization approach is robust enough and my suggestion is beyond the scope of the work, I will accept that.

Reviewer #3: The manuscript by Wang et al. describes a well-conducted and very relevant example of reuse of proteomics datasets available in public repositories. Twenty-three datasets from Pride corresponding to 211 samples originating from 34 tissues across 14 organs and including mouse and rat strains were used. First, they have elegantly extracted comparative protein expression maps between different tissues/organs of a given species to propose baseline protein expression profiles before deducing organ-specific enriched biological processes and pathways. The authors have previously applied an equivalent strategy on human baseline datasets coming from 32 different organs (Prakash A, et al. 2021, bioRxiv) and to compare human cell lines and tumour samples (Jarnuczak AF, et al. 2021, Sci Data). The originality of the present work relies in the cross-species comparisons that were further added. Indeed, the authors also conducted orthologs analyses to compare protein expression profiles of different tissue/organ types across mouse, rat and human samples. Finally, the output of the study has been integrated into the Expression Atlas, which is a nice way to make the work widely available.

As a specific remark, it is a shame that the current status of annotation/metadata availability of public datasets still requires, prior and fastidious, thorough manual cleaning/reannotation of the datasets before they can be reused for such a study. The authors should even more clearly highlight this shortcoming which constitutes a real brake to this type of studies and more generally to the re-use of public datasets.

This work is worth being published in a journal like PLOS Computational Biology as this strategy can, and should be, increasingly applied to a wide range of available –omics, and in particular, proteomics datasets.

I have only two major comments that should be adressed in a revised version of the manuscript :

- Instead of using finely extracted quantitative data from MaxQuant and then proceeding to a "coarse" binning, the authors should conduct the same analysis directly on spectral counting data (ex. length-normalised unique peptide counts). It would be very interesting to show whether/or not this has an impact on the results.

- The authors should correlate their results achieved with proteomics data with antibody-based data extracted from Protein Atlas. This later inclusion would provide an added-value to the work.

**Have the authors made all data and (if applicable) computational code underlying the findings in their manuscript fully available?**

Reviewer #1: Yes

Reviewer #2: Yes

Reviewer #3: Yes

PLOS authors have the option to publish the peer review history of their article (what does this mean?). If published, this will include your full peer review and any attached files.

Reviewer #1: No

Reviewer #2: **Yes: **Hyungwon Choi

Reviewer #3: No
---

## [Decision Letter · Decision Letter 1]

7 Apr 2022

Dear Dr. Prakash,

Thank you very much for submitting your manuscript "Integrated view and comparative analysis of baseline protein expression in mouse and rat tissues" for consideration at PLOS Computational Biology. As with all papers reviewed by the journal, your manuscript was reviewed by members of the editorial board and by several independent reviewers. The reviewers appreciated the attention to an important topic. Based on the reviews, we are likely to accept this manuscript for publication, providing that you modify the manuscript according to the review recommendations.

Sincerely,

Zhaolei Zhang

Associate Editor

PLOS Computational Biology

Ilya Ioshikhes

Deputy Editor

PLOS Computational Biology

[LINK]

Reviewer's Responses to Questions

**Comments to the Authors:**

Reviewer #1: Thank authors for addressing my questions. I still have questions regarding Question 2 and Question 5.

For Question 2, from my personal experience, spectral counts and ion intensity are highly correlated (also documented in many papers), there is not much difference in practice. Regarding PaxDb paper, I think they applied more interesting integration method to weigh each dataset and peptides and then benchmark using protein-protein interaction information. I think their work is quite innovative and interesting. It is hard to convince me this dataset has better quality or more useful than PaxDb.

For Question 5, I would suggest the authors to remove the pathway analysis section from the paper. The result is not very useful as detecting many enriched pathways is expected by the authors’ setting, unless the organ/tissue specific pathways can be reported, which would be very interesting results.

I have no further questions.

Reviewer #2: Authors have addressed my major concerns in the first review, and the manuscript has more substance than the initial submission. However, I strongly recommend the authors to consider rewriting the abstract, focusing on the key results rather than stating what was performed, e.g. comparison of orthologues and pathway enrichment analysis, etc.

**Have the authors made all data and (if applicable) computational code underlying the findings in their manuscript fully available?**

Reviewer #1: None

Reviewer #2: Yes

PLOS authors have the option to publish the peer review history of their article (what does this mean?). If published, this will include your full peer review and any attached files.

Reviewer #1: No

Reviewer #2: **Yes: **Hyungwon Choi

Figure Files:

Data Requirements:

Reproducibility:

References:

---

## [Decision Letter · Decision Letter 2]

5 May 2022

Dear Dr. Prakash,

We are pleased to inform you that your manuscript 'Integrated view and comparative analysis of baseline protein expression in mouse and rat tissues' has been provisionally accepted for publication in PLOS Computational Biology.

Best regards,

Zhaolei Zhang

Associate Editor

PLOS Computational Biology

Ilya Ioshikhes

Deputy Editor

PLOS Computational Biology

Reviewer's Responses to Questions

**Comments to the Authors:**

Reviewer #1: For Q2, the easiest way is to do the same benchmark analysis as PaxDB paper does and show the performance is better than PaxDB.

For Q5, I don't agree with the authors.

I have no further comments.

**Have the authors made all data and (if applicable) computational code underlying the findings in their manuscript fully available?**

Reviewer #1: Yes

PLOS authors have the option to publish the peer review history of their article (what does this mean?). If published, this will include your full peer review and any attached files.

Reviewer #1: No

---

## [Editor Report · Acceptance letter]

8 Jun 2022

PCOMPBIOL-D-21-02288R2 

Integrated view and comparative analysis of baseline protein expression in mouse and rat tissues

Dear Dr Prakash,

I am pleased to inform you that your manuscript has been formally accepted for publication in PLOS Computational Biology. Your manuscript is now with our production department and you will be notified of the publication date in due course.

With kind regards,

Olena Szabo
